# Three-Dimensional Spheroid Configurations and Cellular Metabolic Properties of Oral Squamous Carcinomas Are Possible Pharmacological and Pathological Indicators

**DOI:** 10.3390/cancers15102793

**Published:** 2023-05-17

**Authors:** Sho Miyamoto, Nami Nishikiori, Tatsuya Sato, Megumi Watanabe, Araya Umetsu, Yuri Tsugeno, Fumihito Hikage, Takashi Sasaya, Hirotaka Kato, Kazuhiro Ogi, Masato Furuhashi, Hiroshi Ohguro, Akihiro Miyazaki

**Affiliations:** 1Departments of Oral Surgery, Sapporo Medical University, S1W17, Chuo-ku, Spporo 060-8556, Japan; miyamoto.sho@sapmed.ac.jp (S.M.); sasaya338@sapmed.ac.jp (T.S.); hirotakato0303@gmail.com (H.K.); ogi@sapmed.ac.jp (K.O.); 2Departments of Ophthalmology, Sapporo Medical University, S1W17, Chuo-ku, Spporo 060-8556, Japan; nami076@yahoo.co.jp (N.N.); watanabe@sapmed.ac.jp (M.W.); araya.umetsu@sapmed.ac.jp (A.U.); yuri.tsugeno@gmail.com (Y.T.); fuhika@gmail.com (F.H.); ooguro@sapmed.ac.jp (H.O.); 3Departments of Cardiovascular, Renal and Metabolic Medicine, Sapporo Medical University, S1W17, Chuo-ku, Spporo 060-8556, Japan; satatsu.bear@gmail.com (T.S.); mfuruhas@gmail.com (M.F.); 4Departments of Cellular Physiology and Signal Transduction, Sapporo Medical University, S1W17, Chuo-ku, Spporo 060-8556, Japan

**Keywords:** three-dimensional spheroid culture, oral squamous carcinoma, seahorse bioanalyzer, KRAS, SOX2

## Abstract

**Simple Summary:**

In the current study, to elucidate the clinicopathological significance of the appearance of in vitro three-dimensional (3D) spheroid models of oral malignant tumors, pathologically, three different squamous cell carcinoma (OSCC) cell lines and one oral adenosquamous carcinoma (OASC) cell line were included. Using these, the biological measurements of the two-dimensional (2D) cultured cells through the cellular metabolic analysis of a Seahorse bio-analyzer and the cytotoxicity of cisplatin (CDDP) resulted in the appearance of their 3D spheroids by phase contrast microscopy and others. In Seahorse real-time cellular metabolic analysis, ATP-linked respiration and proton leaking were significantly different among the four cell lines, and the malignant stages of these cultures were significantly associated with increased ATP-linked respiration and decreased proton leakage. Alternatively, the appearances of these 3D spheroids were also significantly diverse, and the increasing order of their differences was identical to the increased efficacies of CDDP-induced cytotoxicity. The current obtained findings indicate that cellular metabolic functions and 3D spheroid architectures may be valuable and useful indicators when estimating the pathological and drug-sensitive aspects of OSCC and OASC malignancies.

**Abstract:**

The objective of the current study was to elucidate the clinicopathological significance and appearance of in vitro three-dimension (3D) spheroid models of oral malignant tumors that were prepared from four pathologically different squamous cell carcinoma (OSCC; low-grade; SSYP and MO-1000, intermediate-grade; LEM2) and oral adenosquamous carcinoma (OASC; high-grade; Mesimo) obtained from patients with different malignant stages. To characterize the biological significance of these cell lines themselves, two-dimensional (2D) cultured cells were subjected to cellular metabolic analysis by a Seahorse bioanalyzer alongside the measurement of the cytotoxicity of cisplatin (CDDP). The appearance of their 3D spheroids was then observed by phase contrast microscopy, and both 2D and 3D cultured cells were subject to trypsin digestion and qPCR analysis of factors related to oncogenic signaling and other related analyses. ATP-linked respiration and proton leaking were significantly different among the four cell lines, and the malignant stages of these cultures were significantly associated with increased ATP-linked respiration and decreased proton leakage. Alternatively, the appearances of these 3D spheroids were also significantly diverse among them, and their differences increased in the order of LEM2, MO-1000, SSYP, and Mesimo. Interestingly, these orders were exactly the same in that the efficacies of CDDP-induced cytotoxicity increased in the same order. qPCR analysis indicated that the levels of expression of oncogenic signaling-related factors varied among these four cell lines, and the values for fibronectin and a key regulator of mitochondrial biogenesis, PGC-1α, were prominently elevated in cultures of the worst malignant Mesimo cells. In addition, although 0.25% trypsin-induced destruction was comparable among all four 2D cultured cells, the values for the 3D spheroids were also substantially varied among these cultures. The findings reported herein indicate that cellular metabolic functions and 3D spheroid architectures may be valuable and useful indicators for estimating the pathological and drug-sensitive aspects of OSCC and OASC malignancies.

## 1. Introduction

Oral squamous cell carcinoma (OSCC), the most common malignancy of the oral cavity, is recognized as one of the main causes of mortality worldwide, especially in developing countries [1,2]. Despite the fact that some progress has been made in treatment in addition to the fact that an early diagnosis can be made, presentation with advanced stages of OSCC is not uncommon, and the 5-year survival rate has not significantly improved over the past few decades and is still less than 50% across all stages [3,4,5]. As possible etiological risk factors for causing OSCC, interest has focused on inflammation and the microbiome within the oral cavity, in addition to smoking and alcohol intake, based upon the evidence that several inflammatory mediators, salivary proteins, and oral microbiota have been identified as pathologic factors that are shared between oral inflammatory diseases, such as periodontitis and OSCC [6,7,8]. Regarding this standard therapeutic strategy, primary surgical resection can be performed with or without postoperative adjuvant therapies such as radiation and/or chemotherapy to improve survival statistics [9], and survival rates have become markedly higher in patients with OSCC that were diagnosed and treated in earlier stages [8]. In addition, to obtain better therapeutic statistics for OSCC, a further understanding of the molecular pathology of oral carcinogenesis using biologically relevant models is required to facilitate the identification of potential biomarkers of OSCC carcinogenesis as well as the testing of several anti-tumor drugs to develop better chemotherapy.

For this purpose, initially, conventional two-dimensional (2D) planar cell culture methods have been used as in vitro disease models. Subsequently, however, it was recognized that the nature of these 2D cultured models was distinct from those of in vivo physiological states. This conclusion is based on several differences in changes of several gene expression profiles, intercellular binding, extracellular matrix protein interactions, and others between normal or cancerous tissues [10,11]. By contrast, since in vivo OSCC cells are located within three-dimensional (3D) solid tumors, they are expected to be exposed to various gradients of several biological molecules and signals. This suggests that more physiologically relevant 3D cell cultures in vitro models that replicate such in vivo OSCC environments are needed for the successful screening of new antitumor drugs, the more accurate evaluation of their efficacy as well as suitable dosages of anticancer agents [12,13,14,15,16]. Among the several types of in vitro 3D cell culture models, a simpler form called a 3D spheroid that can be obtained in a nonadherent surface manner has been the most frequently used model [17,18]. We recently and independently succeeded at producing in vitro 3D spheroid models that replicate several disease states using non-cancerous cells including 3T3-L1 preadipocytes, human orbital fibroblasts (HOF) [19], human trabecular meshwork (HTM) cells [20], human conjunctival fibroblasts (HconF) [21] as well as other cancerous cell lines, including an A549 lung adenocarcinoma cell [22], and various malignant melanoma cell lines [23]. Based upon these collective studies, we concluded that the biological aspects of 3D spheroids were significantly different from those of 2D cultured cells even though both are cultured under the exact same conditions, except that different culture plates are used [24].

In addition, it was also found that the shapes of 3D spheroids were dependent on the origin of the cells. Globe-shaped or non-globe-shaped 3D spheroids were obtained from non-cancerous cells [19,20,21] or cancerous cells, including malignant melanoma (MM) [25] and others, respectively. In our recent study, we found that the configurations of 3D spheroids that were generated from five different MM cell lines (SK-mel-24, MM418, A375, WM266-4, and SM2-1) and dabrafenib and trametinib resistant A375 (A375DT) were substantially deformed and that these levels were correlated with cellular metabolic functions, as measured by a Seahorse Bioanalyzer. Furthermore, RNA sequencing analyses of the two distinct cell lines, WM266-4 and SK-mel-24, indicated that differences in the appearance of 3D cultures suggested that KRAS and SOX2 might be potential master regulatory genes for inducing diverse 3D configurations. These collective findings rationally suggested that a 3D spheroid configuration has the potential to serve as an indicator and evaluate the pathophysiological activities of MM. We, therefore, concluded that the characterization of such in vitro 3D spheroid models might be useful for the pathophysiological investigation of other malignant tumors, including OSCC. 

Therefore, in the current study, to elucidate the clinicopathological significance of the in vitro 3D spheroid model of OSCC, 3D spheroids were prepared using four OSCC or oral adenosquamous carcinoma (OASC) cell lines, LEM2, M0-1000, SSYP and Mesimo which were established from patients with clinically different malignant stages, and the morphological and functional aspects of these cultures were compared.

## 2. Materials and Methods

### 2.1. Establishment of Oral Squamous Carcinoma (OSCC) or Oral Adenosquamous Carcinoma (OASC) Cell Lines and Their 2D and 3D Cell Cultures

Three oral squamous carcinoma cells termed SSPY, MO-1000, LEM2, and one type of oral adenosquamous carcinoma cells (OASC) termed MeSimo were uniquely established in our laboratory. In a typical experiment, tumor tissues were minced into 1–2 mm pieces with scissors and placed in a primary culture in Dulbecco’s modified Eagle’s medium/Ham’s F12 medium (Life Technologies, Carlsbad, CA, USA) supplemented with 10% fetal bovine serum (Sigma-Aldrich, St Louis, MO, USA) and 1% penicillin–streptomycin (10,000 U/mL penicillin, 10,000 μg/mL streptomycin; Life Technologies) in a humidified atmosphere containing 5% CO_2_ at 37 °C. When the cells reached 80–90% confluence, they were washed with PBS and detached by treatment with a 0.05% EDTA/trypsin solution for 5–10 min at 37 °C. After centrifugation at 1500 rpm for 5 min at room temperature, the cells were resuspended in a medium and seeded in 100 mm dishes. The passage was serially performed until a cell line was established. The malignancy levels of these cell lines were estimated based on the Yamamoto–Kohama (YK) classification of invasion [26] for their originated patients. SSPY, MO-1000, or LEM2 was established from biopsy specimens with YK-3 well-differentiated OSCC and YK-3 that had moderately differentiated OSCC classified as grade 3 or YK-4C moderately differentiated OSCC, respectively. Alternatively, MeSimo was also established from biopsy specimens with YK-4D poorly differentiated OASC. The clinical and pathological details of these OSCC and OASC cells are summarized in Table 1. The 2D and 3D cell cultures of these four cell lines were processed for 7 days by the methods demonstrated previously using non-cancerous and cancerous cells [19,24,27,28]. 

### 2.2. Real-Time Analysis of the Cellular Metabolic Functions of Various OSCC and OASC Cell Lines

The oxygen consumption rate (OCR) and extracellular acidification rate (ECAR) of 2D-cultured OSCC and OASC cells, SSYP, MO-1000, LEM2, and Mesimo, were measured using a Seahorse XFe96 real-time metabolic analyzer (Agilent Technologies Agilent Technologies, Santa Clara, CA, USA) according to the manufacturer’s instructions. In a typical run, twenty thousand 2D-cultured cells, as described above, were placed in each well of an XFe96 Cell Culture Microplate (Agilent Technologies, #103794-100) the day before the assay and were incubated at 37 °C. On the day of the assay, the culture medium was replaced with a Seahorse XF DMEM assay medium (pH 7.4, Agilent Technologies, #103575-100) containing 5.5 mM glucose, 2.0 mM glutamine, and 1.0 mM sodium pyruvate). The assay plates were then incubated in a CO_2_-free incubator at 37 °C for 1 h prior to the measurements. OCR and ECAR were simultaneously measured in an XFe96 extracellular flux analyzer at the baseline and under the following sequential injections of 2.0 μM oligomycin, 5.0 μM carbonyl cyanide-p-trifluoromethoxyphenylhydrazone (FCCP), a mixture of 1.0 μM rotenone and 1.0 μM antimycin A, and 10 mM 2-deoxyglucose. The values were normalized to the amount of protein, as determined by a BCA protein assay (TaKaRa BCA Protein Assay) in each well after the measurement was complete.

### 2.3. Morphological Analyses of the 3D Spheroids Derived from Various OSCC and OASC Cells

Regarding the morphological aspects of the above-prepared 3D spheroids, their downward and lateral views were observed by phase contrast microscopy, as reported in our previous study [19,24,29]. To compare the degree of deformity levels among the cultures, ratios (%) of the horizontal and vertical configurations of the largest inscribed circle area to the smallest circumscribed circle area in the downward and lateral view of PC images of the 3D spheroids were determined, as reported in our recent study using malignant melanoma cell lines [23].

### 2.4. Assays for Cisplatin-Induced Cytotoxicity

To estimate the cytotoxic effects against cisplatin (CDDP), four different 2D cultured OSCC or OASC cell lines; SSYP, MO-1000, LEM2, and Mesimo (5000 cells/well) were exposed to CDDP at 0, 1.0 × 10^−2^, 1.0 × 10^−1^, 1.0 × 10^0^, 1.0 × 10^1^ or 1.0 × 10^2^ μM concentrations for 72 h. To determine the survival rates, after incubation with 10 μL of a reactive solution of the commercially available kit (Cell Counting Kit-8, Dojindo, Tokyo, Japan) for 2 h, the absorbance at 450 nm was measured using a microplate reader (multimode plate reader EnSpire^®^, PerkinElmer, Waltham, MA, USA).

### 2.5. Assays for Trypsin-Induced Dispersion

To compare the trypsin-induced destruction of the 2D and 3D cultured cells among 4 OSCC and OASC cells, the period required for them to disperse against 0.05% trypsin was determined using phase contrast microscopy.

### 2.6. Other Methods

Quantitative PCR using specific primers (Appendix A) and statistical analyses using the Graph Pad Prism 8 (GraphPad Software, San Diego, CA, USA) were performed as demonstrated in a previous report [24]. For the estimation of statistical differences between the study groups, a Student’s *t*-test for the two-group comparison or one-way ANOVA followed by Tukey’s multiple comparison tests was used.

## 3. Results

### 3.1. Cellular Metabolic Aspects of 2D Cultured Four Established OSCC or OASC Cell Lines

In the current study, we established four OSCC or OASC cell lines that originated from patients with pathologically different malignant levels (low-grade; SSYP and MO-1000, intermediate-grade; LEM2, and high-grade; Mesimo, Table 1). To initially characterize their biological natures, cellular metabolic functions were studied. The results of real-time cellular metabolic analyses using the Seahorse bioanalyzer are shown in Figure 1. Consistent with the increase in the pathological grade of the OSCC or OASC cell lines, ATP-linked respiration was significantly increased, and proton leakage was significantly decreased (Figure 1A,B). This finding indicates that OSCC or OASC cell lines with a higher pathological malignancy consumed oxygen more efficiently to produce ATP. Alternatively, OCR levels were decreased in response to FCCP in SSYP and MO-1000, which were less malignant cells (Figure 1A), indicating that the extent of proton charging was low in the mitochondrial electron transfer system in these cells, presumably due to an impaired mitochondrial function. Non-mitochondrial oxygen consumption was also markedly higher in these cells, suggesting that oxygen may be used not for ATP production but, rather, for the generation of reactive oxygen species (ROS) or the production of gases such as NO, which may be related to mitochondrial dysfunction. On the other hand, the trend for glycolytic capacity was higher in the other three groups compared to SSYP (Figure 1C,D). In addition, non-glycolytic acidification also decreased with the increasing pathological grade of the OSCC cell lines (Figure 1E). This finding may be associated with proton releases independent of glycolysis, such as glutaminolysis or the increased breakdown of ATP to ADP.

### 3.2. Cytotoxicity of 2D Cultured Four Established OSCC or OASC Cell Lines against Cisplatin (CDDP)

To elucidate additional biological diversity among the four established OSCC or OASC cell lines, the levels of CDDP-induced cytotoxicity were determined. As shown in Figure 2, the differences among them were dramatic in that they were significantly lower or higher in the high-grade cell line, Mesimo, or the intermediate-grade cell line, LEM2, respectively, compared with the low-grade cell lines, SSYP and MO-1000.

### 3.3. D Spheroid Configurations of Four Established OSCC or OASC Cell Lines

To examine the 3D spheroid configurations obtained from these four different cell lines, horizontal and lateral PC images of these cultures were obtained, and the results indicated that the configurations were significantly diverse among these cell lines (Figure 3). To evaluate these differences, the deformity levels of the horizontal and vertical configurations were determined using the downward and lateral view of PC images in the 3D spheroids. As shown in Figure 3, the ratio (%) of the largest inscribed circle area to the smallest circumscribed circle area in the horizontal and vertical images were LEM2 (88.9% and 95.6%), MO-1000 (80.0% and 60.7%), SSYP (77.9% and 62.9%) and Mesimo (69.4% and 50.0%), respectively. This result indicates that the degrees of deformity in the spheroids that originated from four different OSCC or OASC cells were increased in the order LEM2, M0-1000, SSYP, and Mesimo, which is the same order as the decreasing sensitivities to CDDP among the cultures (Figure 2).

### 3.4. Trypsin-Induced Dispersion of 2D and 3D Cultures of Four Established OSCC or OASC Cell Lines

To elucidate the intercellular binding properties of the four 2D and 3D cultured OSCC or OASC cell lines, their trypsin-induced dispersion was compared. As shown in Figure 4, a 0.25% trypsin solution-induced the destruction of all four 2D cultured cells and reached completion within 5–10 min. By contrast, the times for the 3D spheroids were much longer, though the times required for MO-1000 (2 h) and Mesimo (3 h) were quite similar to and faster or slower compared with LEM2 (33 h) or SSYP (1 h), respectively.

### 3.5. The mRNA Expression of Possibly Related Molecules within the Biological Diversities among 2D and 3D Cultures of the Four Established OSCC or OASC Cell Lines

To elucidate the possible underlying molecular mechanisms that were responsible for inducing these diversities among the four different OSCC or OASC cells, the mRNA expressions of several possibly related molecules of oncogenic signaling-related factors including KRAS, SOX2, and MITF, and major ECMs including COL4, COL6 and FN, and PGC-1α, were investigated as a key regulator of mitochondrial metabolism. As shown in Figure 5, the gene expression levels of (1) the oncogenic signaling related factors, KRAS, SOX2, and MITF, were higher in MO-1000 or lower in LEM2, respectively, (2) major ECM proteins, such as COL4, were significantly higher in MO-1000 and Mesimo, COL6 was higher in LEM2, and FN1 was significantly higher in Mesimo, and (3) a key regulator of mitochondrial biogenesis, PGC-1α, was significantly higher in Mesimo. The profiles for the gene expressions of KRAS, MITF and COL4 were similar between MO-1000 and Mesimo. In addition, higher levels of the mRNA expressions of PGC-1α were consistent with the increased mitochondrial functions determined by a Seahorse real-time cellular metabolic analysis, as observed above.

## 4. Discussion

In vitro 3D spheroids, or organoid models, have attracted considerable interest for use because they are more physiologically relevant in terms of replicating the microenvironment of malignant tumors compared with the traditional 2D planar cell culture models [30,31]. However, in the case of OSCC and OASC, these issues have been poorly investigated [32]. It is particularly noteworthy that studies focusing on the pathological and pharmacological significance of the appearance of 3D spheroids remain quite limited. In recent studies, we demonstrated that the appearance of 3D spheroids that had originated from non-cancerous cells and cancerous cells were globe-shaped or non-globe-shaped, respectively. In addition, in four different malignant melanoma (MM) cell lines, we also found that the degree of deformity in their 3D spheroids was significantly diverse among these lines and was closely correlated with their metabolic functions [23]. This observation prompted us to speculate that such differences in the levels of the appearance of 3D cancerous spheroids may be correlated with some clinical aspects, such as the malignancy grade, drug sensitivity against anti-tumor agents, and others. Therefore, to elucidate which of these clinical aspects of OSCC were correlated with the appearance of corresponding 3D spheroids, four OSCC or OASC cell lines that originated from patients with pathologically different malignant levels (low-grade; SSYP and MO-1000, intermediate-grade; LEM2, and high-grade; Mesimo, Table 1) were established and used in the present study. The following results were obtained; (1) the pathological malignancy of these OSCC and OASC cell lines was closely related to the level of production for the ATP ion in their mitochondria, as summarized in Figure 1E, and (2) the degree of pathological malignancy and pharmacological anti-tumor drug sensitivities corresponded well with the cellular metabolic functions and the appearance of 3D spheroids, respectively, in the OSCC and OASC. These collective results indicated that cellular metabolic functions and 3D spheroid architectures might be valuable indicators for the evaluation of the pharmacological and pathological aspects of OSCC and OASC malignancies.

In the cellular metabolic analyses, we found significant differences among the four OSCC or OASC cell lines, with a higher oxygen utilization for ATP production in the mitochondria as the pathological malignancy progressed. In Mesimo, the most pathologically malignant among the four OSCC or OASC cell lines, the mitochondrial respiratory function, was enhanced with the increased gene expression of PGC-1α, whereas, in the lesser malignant cells (SSYP and MO-1000), proton leakage increased and an FCCP-induced increase in oxygen consumption was impaired, suggesting that these lesser malignant cells damaged the mitochondria. Glycolytic capacity was relatively low in low-grade SSYP cells but remained at high levels in all four OSCC cell lines. These associations between malignancy and cellular metabolism may reflect the fact that these cells have a high energy demand to continuously proliferate in squamous carcinoma cells. Interestingly, Mesimo, which had the highest mitochondrial respiratory capacity, was the most resistant to the anticancer agent CDDP. We (and others) have reported the possibility that enhanced mitochondrial respiration or mitochondrial adaptation can be associated with a variety of anticancer drug resistance in cancer cells [23,33]. Thus, strategies for targeting aberrantly enhanced mitochondrial respiration have the potential to develop strategies for the treatment of high-grade squamous cells or chemotherapy-resistant OSCC. Several prior studies have demonstrated that metformin, an oral hypoglycemic agent that inhibits mitochondrial respiration, is a promising drug for the treatment of anticancer agent-resistant cancers [34,35]. On the other hand, the results in the present study using OSCC cell lines were actually contrary to our previous findings that an increased deformity of 3D spheroids was associated with decreased mitochondrial respiration in melanoma cell lines [23]. Although the precise reason for this discrepancy currently remains unidentified, we speculated that the originated cell type of malignant tumors, such as epithelial or non-epithelial, may be related. In fact, the ball-shaped 3D spheroid appearance of LEM2 quite resembled those obtained from non-cancerous cells, although the 3D spheroids of MM cell lines were significantly deformed [23]. Nevertheless, assessing the association between the morphological characteristics of 3D spheroids and cellular metabolism in various cancer cells would be useful in terms of characterizing the cellular properties of cancer cells and could lead to a better understanding of their pathophysiology as well as possible new therapeutic approaches.

ECM proteins are important molecular components that are involved in cellular architectures and have several signaling properties [36,37,38]. Thus, the composition of these ECM compositions within a tumor type is a useful predictor for determining the various pathological aspects of these tumors. In fact, the composition profile of these ECM molecules contains not only similar characteristics of the organ from which they originated but also tumor-specific cellular composition, in which cancer-associated fibroblasts are known to play a pivotal role in the production of tumor-specific ECM molecules [39]. Among several ECM proteins, FN is a major ECM constituent that binds to several cellular adhesion receptors, thereby supporting cell-to-ECM interactions during several biological processes such as development, maintaining tissue homeostasis, wound healing, and others [40,41,42,43,44]. It has been shown that FN expression is elevated in many solid tumors and is presumably involved in tumor progression, angiogenesis, metastasis, and chemoresistance since cancerous cells and endothelial cells can produce more FN molecules than those that are secreted from fibroblasts within the originated non-cancerous cells [42,45,46,47]. Alternatively, it is well known that COL 4 is an important ECM protein that functions as a predictor of the aggressiveness of OSCC metastatic cells since its expression pattern within the basement membrane is greatly affected by the cervical lymph node metastasis of OSCC [48]. In addition, COL6, also a major ECM protein, formed a discontinuous network of beaded microfilaments that interacted with other ECM molecules, thus contributing to their cellular structural support [49] and were involved in influencing the tumor microenvironment [50,51]. Within malignant tumors, it is also known that COL6 is expressed in high levels and promotes the growth and metastasis of cancerous cells to accelerate their progression by affecting other microenvironmental compartments [51,52]. In the current study, the mRNA expression of major ECMs, as described above, was substantially different among the OSCC and OASC cell lines, and this diversity was significantly correlated with biological aspects such as intercellular binding properties, as evaluated by the resistance to trypsin-induced destruction. In this context, a prominent higher expression of COL6 was observed in the most resistant LEM2; the expressions of both COL4 and COL6 were elevated in the moderately resistant MO-1000 and Mesimo, and the expressions of both COL4 and COL6 were insufficient in the lowest resistant SSYP. In addition, the prominent higher expression of FN, which is known to be related to malignancy, was observed in the high-grade malignant cell line, Mesimo.

However, current conclusions remain speculative at present, and the following study limitations need to be investigated regarding unidentified issues, including the relationship between the appearance of the 3D spheroids and several additional clinical, physiological, and pharmacological aspects. Thus, to elucidate the additional significance of the in vitro 3D spheroid model regarding the pathogenesis of oral malignant tumors in more detail, additional investigations with the objective of identifying additional regulatory factors and mechanisms using approaches such as RNA sequencing and others will be our next project.

## 5. Conclusions

In the current study, we described the clinicopathological significance of newly developed in vitro 3D spheroid models of oral malignant tumors using pathologically different OSCC and OASC cell lines. Several biological measurements, including a real-time cellular metabolic analysis, the cytotoxicity of cisplatin (CDDP), the appearance of their 3D spheroids, and others resulted in cellular metabolic functions and 3D spheroid appearance, which could become pivotal and useful bio-indicators for evaluating clinicopathological significance; that is, evaluations could be conducted on the pathological and drug-sensitive aspects of OSCC and OASC malignancies. Therefore, our currently developed in vitro 3D OSCC spheroid model may be a useful methodology for related research investigations.

## Figures and Tables

**Figure 1 cancers-15-02793-f001:**
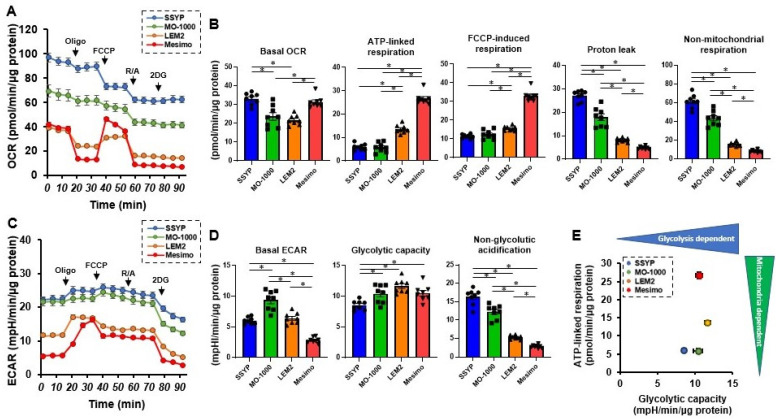
Measurement of the mitochondrial and glycolytic functions of four OSCC and OASC cell lines. The Seahorse real-time cellular metabolic functions of the four different 2D cultured OSCC or OASC cell lines, SSYP, MO-1000, LEM2, and Mesimo, were measured. Panel (**A**); Fluctuations in oxygen consumption rate (OCR) upon subsequent supplementation with a complex V inhibitor oligomycin (Oligo), a protonphore carbonyl cyanide-p-trifluoromethoxyphenylhydrazone (FCCP), complex I/III inhibitors, rotenone/antimycin A (R/A), and a hexokinase inhibitor, 2-deoxy-Dglucose (2DG). Panel (**B**); Key parameters of mitochondrial respiration including (1) basal OCR; the baseline OCR—the OCR with R/A, (2) ATP-linked respiration; the baseline OCR—the OCR with Oligo, (3) Proton leakage; the OCR with Oligo—the OCR with R/A, (4) FCCP-induced respiration; the OCR with FCCP—the OCR with R/A, (5) Non-mitochondrial respiration; the OCR with R/A. Panel (**C**); Fluctuation of extracellular acidification rate (ECAR) upon subsequent supplementation with Oligo, FCCP, R/A, and 2DG. Panel (**D**); Key parameters of glycolytic function including (1) Basal ECAR; the baseline ECAR—ECAR with 2DG, (2) Glycolytic capacity; the ECAR with Oligo—ECAR with 2DG, (3) Non-glycolytic acidification; the ECAR with 2DG. Panel (**E**); Figure showing the relationship between ATP-linked respiration and glycolytic capacity for each of the four OSCC or OASC cell lines. All experiments were performed using fresh preparations (n = 8). * *p* < 0.05. (One-way ANOVA followed by Tukey’s multiple comparison test).

**Figure 2 cancers-15-02793-f002:**
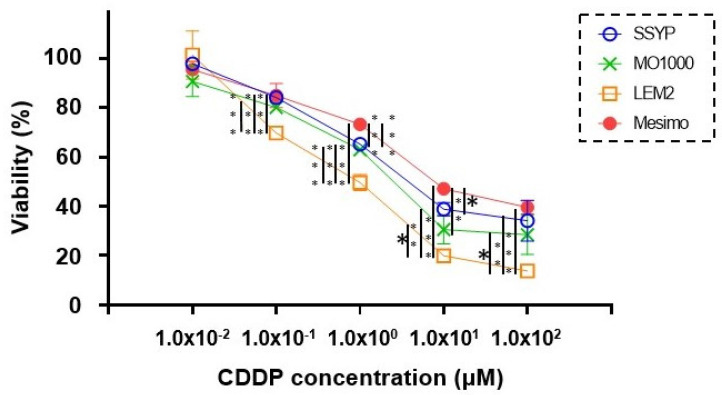
Cisplatin (CDDP)−induced cytotoxicity toward 4 different OSCC or OASC cell lines. To evaluate cisplatin (CDDP)−induced cytotoxicity, assays of the 2D cultures of four OSCC or OASC cell lines; LEM2, MO-1000, SSYP, and Mesimo, survival living cells detected using a CCK8 kit were plotted (n = 3). * *p* < 0.05, ** *p* < 0.01, *** *p* < 0.001 (Student’s *t*-test).

**Figure 3 cancers-15-02793-f003:**
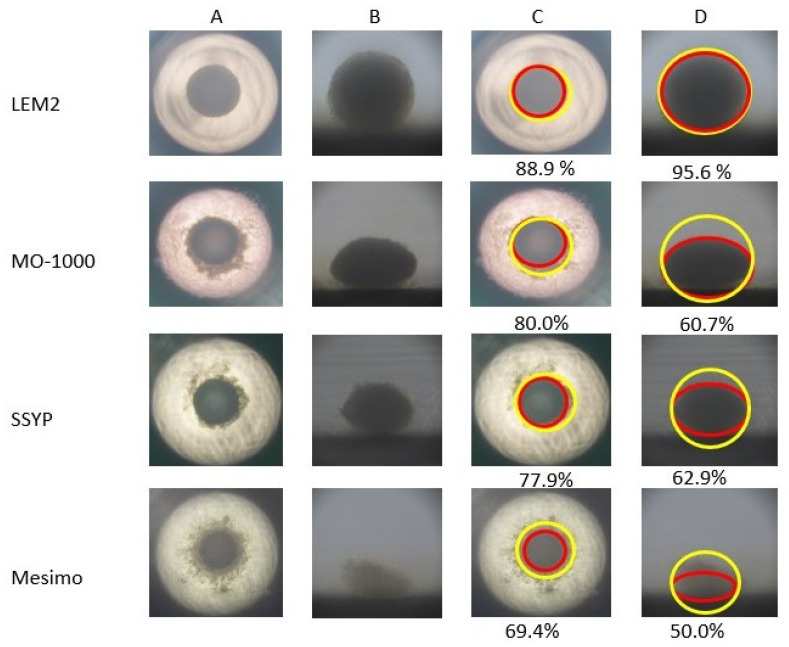
Representative PC (phase contrast microscopy) images of 3D spheroids derived from 4 different OSCC or OASC cell lines. Representative downward (**A**,**C**) or lateral (**B**,**D**) PC images of the 3D spheroids obtained from the four different 2D cultured OSCC or OASC cell lines; LEM2, MO-1000, SSYP, and Mesimo. The ratio (%) of the largest inscribed circle areas (yellow) and smallest circumscribed circle areas (red) is shown below each image in panels (**C**,**D**). Scale bar; 100 μm.

**Figure 4 cancers-15-02793-f004:**
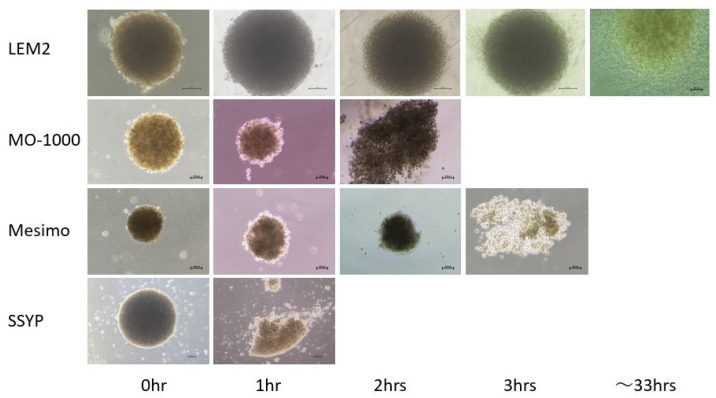
Trypsin digestion of 3D-cultured 4 different OSCC or OASC cell lines. Three-dimensional cultures of 4 different OSCC or OASC cell lines, SSYP, MO-1000, LEM2, and Mesimo, were each treated with 0.025% trypsin until their dispersion was complete. Representative phase contrast microscopy images of 3D spheroids are shown (scale bar; 100 μm). Experiments were repeated in triplicate using fresh preparations (3D; n = 10 spheroids each).

**Figure 5 cancers-15-02793-f005:**
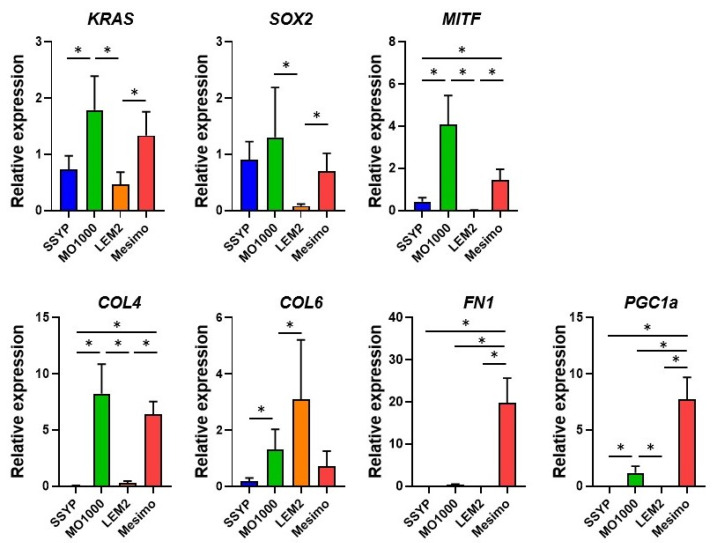
qPCR analysis of oncogenic signaling-related factors (*KRAS, SOX2,* and *MITF*), ECMs (*COL4, COL6,* and *FN*), and a key regulator of mitochondrial biogenesis (*PGC-1α*) among 4 different 2D cultured OSCC cell lines. Among the 2D cultured cells obtained from 4 different OSCC or OASC cell lines, SSYP, MO-1000, LEM2, and Mesimo, the mRNA expressions of *KRAS, SOX2, MITF, COL4, COL6, FN, αSMA,* and *PGC1a,* were evaluated by a qPCR procedure. All experiments were performed in triplicate, each of which used freshly prepared 2D (n = 3) in each experimental condition. * *p* < 0.05.

**Table 1 cancers-15-02793-t001:** Clinical and pathological details of the 4 different OSCC or OASC cell lines; SSYP, MO-1000, LEM2 and Mesimo.

Cell Line	Tumor Site	Histological Type	TNM	Stage	Pathological Grade	YK Classification of Invasion
SSYP	Oral floor	OSCC	T4aN2cM0	IVA	1 (well differentiated)	3
MO-1000	Gingiva	OSCC	T4aN2cM0	IVA	2 (moderately differentiated)	3
LEM	Gingiva	OSCC	T2N0M0	II	2 (moderately differentiated)	4C
Mesimo	Palate	OAdSCC	T3N3bM0	IVB	3 (poorly differentiated)	4D

YK classification of invasion: Yamamoto–Kohama classification of invasion, OSCC: oral squamous cell carcinoma, OASC: oral adenosquamous carcinoma.

## Data Availability

The data can be shared on request.

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
