# Peer review of "Three-Dimensional Spheroid Configurations and Cellular Metabolic Properties of Oral Squamous Carcinomas Are Possible Pharmacological and Pathological Indicators"

_cancers, 2023, doi:10.3390/cancers15102793_

Round 1
Reviewer 1 Report
3D spheroid configurations and cellular metabolic properties 2 of oral squamous carcinomas are possible pharmacological and 3 pathological indictors, respectively
This is an original article that four different types of cell lines originated from oral squamous cancer patients and evaluated their characteristics with mitochondrial and glycolytic functions, cytotoxicity treated with cisplatin, three-dimensional morphological features including destruction by trypsin treatment and oncogenic signalling. I think their results will be attractive and offer a new potency for the cancer therapy strategy resulting in big interest from clinicians and researchers in pharmaceutical and oncological areas. However, it could be deplorable that the structure of the manuscript writing needs to meet the criteria for submission to a certain journal, and the authors should consider what they want to state in each section. They should remember the definition of each introduction section, material and methods, results, discussion, and figure legend. They should not state methodology in the result section or figure legend or mention something related to the discussion part in the result section. Their mix-mingled configuration of the manuscript may be chaotic.
There are several points that I would offer for improving this paper before publication. If the authors consider my suggestion, it would be very appreciated.
- “indictors” in the title appears to be strange. It should be “indicators”, right?
- The first author is usually only one. Two authors are equally devoted to the experiments occasionally, but it is the first time I have met three authors equally devoted to the experiments. There may be a complicated situation. However, three equally devoted authors are unusual. Please clarify who was the most devoted one.
- The introduction is alright but slightly redundant. Can you brush up?
- They should state “agreement of ethical committee”, “COI”, and “patients’ consent” in the manuscript.
- Line 117-118; they used four cell lines from oral cancer patients-SSPY, MO-1000, LEM2 and Mesimo. What are the origins of their name? If the cell lines were established in your own laboratory, you should try to exclude the patient’s personal information.
- Line 127-129; If four cell lines were established in your laboratory, you should summarize how to establish them as cell lines. Showing your previous publications as references may not be sufficient because cell line establishment is not an easy technique, and many authors are interested in how to do it.
- Line 149; “~ as above, their, mean size”- punctuation between “their” and “mean” is not necessary.
- Line 152; you should add the summary of the methods. Offering more than one reference (19) is required.
- Line 153-159; the methods for the cellular reaction of cisplatin and trypsin should be divided into two subsections.
- Line 167-176; This part appears to be discussion. “In our recent studies” is not adequate for the first word of results.
- You should divide the result section into adequately separated subsections.
- Line 188; “Interestingly” is a word of opinion and is very subjective. It would be best to try to be objective when you state experimental results. You can be slightly subjective in the discussion part.
- Line 193; “but, rather, were”- I do not understand this.
- Line 194; “suggesting”- this word is totally inadequate for the result section.
- Line 205-208; this should be stated in the discussion part.
- Line 231-236; this should be stated in the material and methods section. You should state what you can know from the figure in the figure legends.
- Line 240, “as reported in our recent study using MM cell lines [23],” should be stated in the material and methods part.
- Line 243; “the results indicated” should be stated in the discussion part.
- Line 243-245; I do not understand this sentence due to wrong grammar.
- Line 245; “Quite interestingly” is not adequate for manuscript writing. Be objective.
- Line 248-251; You should state this in the discussion part.
- Line 256-259; You should move this part to the material and methods.
- Line 266-268; you should learn to use colon and semi-colon. I do not understand this sentence.
- Line 271: “suggesting” should be mentioned in the discussion part.
- In Figure 4, where is the result of “a-SMA”?
- Line 281; “In fact, although” is not understandable.
- Line 288-290 should be stated in the discussion part.
- Moreover, describe the result of 2D and 3D cell culture and characteristic reaction to cisplatin and trypsin here.
- Line 293-296; Is this part for the figure legend of Figure 5 or the text part?
- Discussion is relatively redundant. Line 301-323 may not be necessary and may make readers confused.
- Line 311; “siRNA”? It would be best to distinguish between upper or lower letters carefully.
- Throughout the whole of the manuscript, authors appear to prefer to set conjunction at the beginning of the sentence, which may result in a strange impression. “Indeed” and repetitive “In fact” may cause indescribable feelings.
- Line 378-382; This is one of the most essential parts of this manuscript. The difference in the expression level of Col4/Col6 and 3D morphology collapse by trypsin treatment in four cell lines should be more precise.
1. They should be careful to distinguish between upper and lower letters.
2. They like to set inadequate conjunctions at the top of the sentences, which may result in confusion of readers.
Author Response
Dear Editor,
Thank you very much for the constructive comments concerning our manuscript, " 3D spheroid configurations and cellular metabolic properties of oral squamous carcinomas are possible pharmacological and pathological indictors, respectively”. We carefully checked all of the Reviewer comments and prepared a revised version of our paper that takes these comments into account. The changes are listed below. Specific changes within the manuscript are highlighted.
Reviewer 1 comment
3D spheroid configurations and cellular metabolic properties 2 of oral squamous carcinomas are possible pharmacological and 3 pathological indictors, respectively
This is an original article that four different types of cell lines originated from oral squamous cancer patients and evaluated their characteristics with mitochondrial and glycolytic functions, cytotoxicity treated with cisplatin, three-dimensional morphological features including destruction by trypsin treatment and oncogenic signalling. I think their results will be attractive and offer a new potency for the cancer therapy strategy resulting in big interest from clinicians and researchers in pharmaceutical and oncological areas. However, it could be deplorable that the structure of the manuscript writing needs to meet the criteria for submission to a certain journal, and the authors should consider what they want to state in each section. They should remember the definition of each introduction section, material and methods, results, discussion, and figure legend. They should not state methodology in the result section or figure legend or mention something related to the discussion part in the result section. Their mix-mingled configuration of the manuscript may be chaotic.
There are several points that I would offer for improving this paper before publication. If the authors consider my suggestion, it would be very appreciated.
- “indictors” in the title appears to be strange. It should be “indicators”, right?
Answer; Thank you for this comment. As pointed out, those “indictors” in the title was changed to “indicators”.
- The first author is usually only one. Two authors are equally devoted to the experiments occasionally, but it is the first time I have met three authors equally devoted to the experiments. There may be a complicated situation. However, three equally devoted authors are unusual. Please clarify who was the most devoted one.
Answer; Thank you for this comment. In terms of authorship within this study, initially the first three authors, Sho Miyamoto, Nami Nishikiori and Tatsuya Sato mainly contributed to complete this study, and thus we wish to specify that these three authors equally contributed to this study. Nevertheless, as suggested, equal contribution of three authors may be vary rare. Evidently, Sho Miyamoto and Nami Nishikiori performed almost the same volume of this investigation, while Tatsuya Sato also worked very hard but his contribution may be less than that for the other 2. In this situation, since I can not clarify the most devoted one between them, if possible, I ask that you permit the equally contribution of two authors, Sho Miyamoto and Nami Nishikiori.
- The introduction is alright but slightly redundant. Can you brush up?
Answer; Thank you for this comment. As suggested, we agree that the 2nd paragraph seemed to be redundant. Therefore, those were revised to separate 2 paragraphs, and 3rd paragraph was rewritten; “In addition, it was also found that the shapes of the 3D spheroids were dependent on the origin of the cells. Globe-shaped or non-globe shaped 3D spheroids were obtained from non-cancerous cells [19-21] or cancerous cells including malignant melanoma (MM) [25] and others, respectively. In our recent study, we found that the configurations of 3D spheroids that were generated from five different MM cell lines (SK-mel-24, MM418, A375, WM266-4 and SM2-1) and dabrafenib and trametinib resistant A375 (A375DT) were substantially deformed and that these levels were correlated with cellular metabolic functions, as measured by a Seahorse Bioanalyzer. Furthermore, RNA sequencing analyses of the two distinct cell lines, WM266-4 and SK-mel-24, indicated that the differences in the appearance of the 3D cultures suggested that KRAS and SOX2 might be potential master regulatory genes for inducing those diverse 3D configurations. These collective findings rationally suggest that a 3D spheroid configuration has the potential for serving as an indicator for evaluating the pathophysiological activities of MM. We therefore conclude that the characterization of such in vitro 3D spheroid models may be useful for the pathophysiological investigation of other malignant tumors, including OSCC.” .
- They should state “agreement of ethical committee”, “COI”, and “patients’ consent” in the manuscript.
Answer; Thank you for this comment. As suggested, statements of “agreement of ethical committee”, “COI”, and “patients’ consent” were included in very first of the Method section “The present study, which was conducted at the Sapporo Medical University Hospital, Japan, was approved by the institutional review board (IRB registration number 342-3416) according to the tenets of the Declaration of Helsinki and national laws for the protection of personal data. Informed consent to use surgical specimens in this study was obtained from all subjects (copies of written informed consent in PDF format are attached in the supplemental material).”.
- Line 117-118; they used four cell lines from oral cancer patients-SSPY, MO-1000, LEM2 and Mesimo. What are the origins of their name? If the cell lines were established in your own laboratory, you should try to exclude the patient’s personal information.
Answer; Thank you for this comment. These cell lines were termed after the name of the originator or the date that the cell lines were established. A suggested, the patient’s personal information within Table 1 and method were deleted.
- Line 127-129; If four cell lines were established in your laboratory, you should summarize how to establish them as cell lines. Showing your previous publications as references may not be sufficient because cell line establishment is not an easy technique, and many authors are interested in how to do it.
Answer; Thank you for this comment. As suggested, the details how to establish OSCC cell lines were included in the method; “In a typical experiment, tumor tissues were minced into 1-2-mm pieces with scissors and placed in primary culture in Dulbecco’s modified Eagle’s medium/Ham’s F12 medium (Life Technologies, Carlsbad, CA, USA) supplemented with 10 % fetal bovine serum (Sigma-Aldrich, St Louis, MO, USA) and 1 % penicillin–streptomycin (10,000 U/ml penicillin, 10,000 μg/ml streptomycin; Life Technologies) in a humidified atmosphere containing 5 % CO2 at 37ËšC. When the cells reached 80-90 % confluence, they were washed with PBS and detached by treatment with a 0.05 % EDTA/trypsin solution for 5-10 min at 37ËšC. After centrifugation at 1,500 rpm for 5 min at room temperature, the cells were resuspended in medium and seeded in 100 mm dishes. The passage was serially performed until a cell line was established.”
- Line 149; “~ as above, their, mean size”- punctuation between “their” and “mean” is not necessary.
Answer; Thank you for this comment. As pointed out, this is a careless mistake regarding their, mean size”- punctuation between “their” and “mean” was corrected.
- Line 152; you should add the summary of the methods. Offering more than one reference (19) is required.
Answer; Thank you for this comment. As suggested, other related references (ref 24, 29) were included.
- Line 153-159; the methods for the cellular reaction of cisplatin and trypsin should be divided into two subsections.
Answer; Thank you for this comment. As suggested, those methods were divided into two subsections.
- Line 167-176; This part appears to be discussion. “In our recent studies” is not adequate for the first word of results.
Answer; Thank you for this comment. As suggested, this information has now been moved to the Discussion.
- You should divide the result section into adequately separated subsections.
Answer; Thank you for such constructive comment. As suggested, result sections were divided using subheadings.
- Line 188; “Interestingly” is a word of opinion and is very subjective. It would be best to try to be objective when you state experimental results. You can be slightly subjective in the discussion part.
Answer; Thank you for this critical comment. As suggested, this sentence was changed to be more objective; “Consistent with the increase in the pathological grade of the OSCC or OAdSCC cell lines, ATP-linked respiration was significantly increased and proton leakage was significantly decreased (Figs. 1A and 1B).”
- Line 193; “but, rather, were”- I do not understand this.
- Line 194; “suggesting”- this word is totally inadequate for the result section.
Answers for #13, 14; Thank you for these comments. As pointed out, those were corrected to avoid infelicity words; “Alternatively, the OCR levels were decreased in response to FCCP in SSYP and MO-1000, which are less malignant cells (Fig 1A), indicating that the extent of proton charging is low in the mitochondrial electron transfer system in these cells, presumably due to impaired mitochondrial function.”.
- Line 205-208; this should be stated in the discussion part.
Answer; Thank you for this comment. As suggested, those were moved to Discussion.
- Line 231-236; this should be stated in the material and methods section. You should state what you can know from the figure in the figure legends.
Answer; Thank you for this comment. As suggested, the corresponding methods and figure legend were revised.
- Line 240, “as reported in our recent study using MM cell lines [23],” should be stated in the material and methods part.
- Line 243; “the results indicated” should be stated in the discussion part.
- Line 243-245; I do not understand this sentence due to wrong grammar.
Answers for #17-19; Thank you for this critical comment. I apologize that the corresponding methods related to the 3D spheroid configuration were in error. Therefore, as suggested, the corresponding method and result were revised; “.
- Line 245; “Quite interestingly” is not adequate for manuscript writing. Be objective.
Answer; Thank you for this comment. As suggested, this sentence was revised to be more objective.
- Line 248-251; You should state this in the discussion part.
Answer; Thank you for this comment. as suggested, those sentences were moved to the Discussion section.
- Line 256-259; You should move this part to the material and methods.
Answer; Thank you for this comment. As suggested, corresponding legend and method were revised.
- Line 266-268; you should learn to use colon and semi-colon. I do not understand this sentence.
Answer; Thank you for this comment. We apologize these careless mistakes, and therefore, those phrase was revised.
- Line 271: “suggesting” should be mentioned in the discussion part.
Answer; Thank you for this comment. As suggested, this information was moved to Discussion.
- In Figure 4, where is the result of “a-SMA”?
Answer; Thank you for this comment. We apologize for this careless mistake to include aSMA, which was not determined, and therefore, this was removed.
- Line 281; “In fact, although” is not understandable.
Answer; Thank you for this comment. We agree that this is a somewhat indirect expression, and thus those were revised.
- Line 288-290 should be stated in the discussion part.
Answer; Thank you for this comment. As suggested, those information were moved to Discussion.
- Moreover, describe the result of 2D and 3D cell culture and characteristic reaction to cisplatin and trypsin here.
Answer; Thank you for this constructive comment. We agree that result construction should be changed. We think that the reaction to cisplatin using 2D cells may be related to the biological aspects of tumor cells themselves, and therefore it would still be better to show the seahorse measurements and cisplatin cytotoxicity, first, and then the 3D spheroid configuration. However, as suggested, trypsin induced reactions should be better than before qPCR analysis, and therefore, results were revised in those orders.
- Line 293-296; Is this part for the figure legend of Figure 5 or the text part?
Answer; Thank you for this comment. I agree that figure 5 (now changed to Fig. 4) were not suitably located. Therefore, those location was changed.
- Discussion is relatively redundant. Line 301-323 may not be necessary and may make readers confused.
Answer; Thank you for this comment. As pointed out, those phrases appear to be overlapped with the information within the introduction, and therefore Introduction (3rd paragraph) and Discussion (1st paragraph) were revised to avoid redundant as much as possible.
- Line 311; “siRNA”? It would be best to distinguish between upper or lower letters carefully.
Answer; Thank you for this comment. I apologize such careless mistake, those was changed.
- Throughout the whole of the manuscript, authors appear to prefer to set conjunction at the beginning of the sentence, which may result in a strange impression. “Indeed” and repetitive “In fact” may cause indescribable feelings.
Answer; Thank you for such professional comment. As suggested such infelicity expression was removed.
- Line 378-382; This is one of the most essential parts of this manuscript. The difference in the expression level of Col4/Col6 and 3D morphology collapse by trypsin treatment in four cell lines should be more precise.
Answer; Thank you for this comment. As suggested, corresponding phrase were revised to be more precise; “In the current study, the mRNA expression of major ECMs, as described above, were substantially different among the OSCC and OAdSCC cell lines and this diversity was significantly correlated with biological aspects such as intercellular binding properties, as evaluated by the resistance to trypsin induced destruction. In this context, a prominent higher expression of COL6 was observed in the most resistant LEM2, the expressions of both COL4 and COL6 were elevated in the moderately resistant MO-1000 and Mesimo and the expressions of both COL4 and COL6 were insufficient in the lowest resistant SSYP. In addition, a prominent higher expression of FN, which is known to be related to malignancy, was observed in the high-grade malignant cell line, Mesimo.”
Comments on the Quality of English Language
- They should be careful to distinguish between upper and lower letters.
- They like to set inadequate conjunctions at the top of the sentences, which may result in confusion of readers.
Answers for these comments; Thank you for these comments. As pointed out, English quality was not well be done, and therefore, manuscript was carefully edited by a native English speaking scientist, professor Milton Feather, and his certification is attached.
Reviewer 2 Report
This study concerns the characterisation of 3D spheroid models in vitro for the pathophysiological study of OSCC. This work is of great interest for future clinical applications.
The following suggestions are given to improve the readability of the article:
The results section is not very smooth in reading. I suggest that the data obtained be made clearer to the audience, e.g. by creating subheadings.
Some abbreviations in the text lack full text (i.e. MM page 2, line 88)
Why did the authors decide not to include a control cell line?
Author Response
Dear Editor,
Thank you very much for the constructive comments concerning our manuscript, " 3D spheroid configurations and cellular metabolic properties of oral squamous carcinomas are possible pharmacological and pathological indictors, respectively”. We carefully checked all of the Reviewer comments and prepared a revised version of our paper that takes these comments into account. The changes are listed below. Specific changes within the manuscript are highlighted.
Reviewer 2 comment
This study concerns the characterisation of 3D spheroid models in vitro for the pathophysiological study of OSCC. This work is of great interest for future clinical applications.
The following suggestions are given to improve the readability of the article:
- The results section is not very smooth in reading. I suggest that the data obtained be made clearer to the audience, e.g. by creating subheadings.
Answer; Thank you for this constructive comment. As suggested, the result sections were revised using sub-headings.
- Some abbreviations in the text lack full text (i.e. MM page 2, line 88)
Answer; Thank you for this comment. As pointed out, we carefully checked to make sure that all abbreviations were fully spelled out.
- Why did the authors decide not to include a control cell line?
Answer; Thank you for this critical comment. The reason why we did not use a normal control cell line was to elucidate the differences between 2D and 3D cultured OSCC spheroids with various malignancy grades. In addition, in our proceeding study using malignant melanoma (MM) cell lines, RNA sequencing analyses indicated that oncogenic factors, KRAS and SOX2 could be master regulators for inducing differences in the biological natures of the 3D MM spheroids. Therefore, if those of 3D OSCC spheroids may be similar to the case of MM, we assume that normal cell line would not be critically important for the present study.